# Is the Neutrophil-to-Lymphocyte Ratio more correlated than C-reactive protein with postoperative complications after major abdominal surgery?

Patrice Forget, Valérie Dinant and Marc De Kock

Department of Anesthesiology, Cliniques universitaires Saint-Luc, Institute of Neuroscience, Université catholique de Louvain, Brussels, Belgium

Corresponding author
Patrice Forget,
forgetpatrice@yahoo.fr

## ABSTRACT

**Background.** The Neutrophil-to-Lymphocyte Ratio (NLR) is an inflammatory marker that has proven usefulness for predicting late complications. Whether it is associated with immediate postoperative complications after abdominal surgery is not known. In this study, we attempted to correlate the NLR and the C-reactive protein (CRP) with postoperative complications rate.

**Methods.** We performed a post-hoc analysis of previously collected data concerning 82 consecutive patients (median age: 62 years, range: 27–80, female/male 32/50) undergoing major abdominal surgeries. For each patient, we recorded preoperative characteristics, the NLR and CRP values, and postoperative complications (between $D + 8$ and $D + 30$) such as infections ($N = 29$), cardiovascular complications ($N = 12$) and other complications ($N = 28$). We performed uni- and multivariate analyses using logistic/linear regression models.

**Results.** Patients with complications did not present a higher preoperative NLR than those without, but a higher ratio at $D + 7$ ($10.73 \pm 9.86$ vs. $4.73 \pm 3.38$ without complication) ($P < 0.001$). In the univariate analysis, the NLR at $D + 7$ was associated with postoperative complications ($P < 0.001$). At $D + 7$, in the multivariate analysis, an increased NLR was associated with more complications ($P < 0.001$), whereas none of the other factors, including CRP, showed any correlation.

**Conclusion.** Postoperative NLR at day 7 after major abdominal surgery is associated with complications during the first postsurgical month, in contrast with the CRP level. The NLR is a simple and interesting parameter in the perioperative period.

## INTRODUCTION

Surgery induces an acute inflammatory response, and many complications can occur in the early postoperative period. Despite advances in surgical techniques and perioperative medicine, excessive surgical stress response can be associated, and even lead, to serious post-operative complications like surgical site infection, sepsis and multiple organ failure (*Tabuchi et al., 2011*). This response can be reflected, at least partially, by

biomarkers that may help the clinician to monitor the patients and to perform early diagnoses. In recent years, the neutrophil-to-lymphocyte ratio (NLR) has been proposed as a simple biological parameter able to stratify the risk of mortality after a major cardiac event and to predict cancer outcome (*Tabuchi et al., 2011*; *Forget et al., 2013*; *Proctor et al., 2011*; *Chua et al., 2011*; *Keizman et al., 2012*; *Huang et al., 2011*; *Kim et al., 2010*; *Ding et al., 2010*; *Chiang et al., 2012*; *Sharaiha et al., 2011*).

In this study, we investigated the potential correlation between the NLR, the C-Reactive Protein (CRP) level and the postoperative complications following major abdominal surgery.

## PATIENTS AND METHODS

### Patients

We performed a post-hoc analysis of previously collected data concerning 82 consecutive patients (median age: 62 years, range: 27–80, female/male: 32/50) undergoing major abdominal surgery included in a previous trial (NCT00816153). After ethical committee approval (CEBHF of the Université catholique de Louvain, Chairperson: Pr J-M Maloteaux) and written informed consent, these patients were prospectively followed. We included adults (>18 years) without any major organ dysfunction, and scheduled for esophagectomy, gastric resection/suture, hepatectomy, pancreatectomy, or intestinal and colorectal surgeries (*Forget, Lois & De Kock, 2010*).

### Data collection

For each patient, a postoperative care team member recorded preoperative characteristics, the NLR and CRP values (preoperative, at days +1, +2 and +7), and delayed postoperative complications (between $D + 8$ and $D + 30$) such as infections, pulmonary embolism, acute myocardial infarction, acute lung injury/acute respiratory distress syndrome, pulmonary edema, arrhythmia, stroke, cardiac arrest, coagulopathy (platelets $< 100,000 \ \mu l^{-1}$, international normalized ratio >2), hepatic dysfunction, upper digestive hemorrhage, leakage of anastomosis, and mortality.

Leukocytes count and CRP were typically included in the routine perioperative evaluation and prospectively registered in a computed database. All venous blood samples were processed in a blood analyzer (Sysmex; TOA Medical Electronics, Kobe, Japan) for the determination of the complete blood cell counts and differential counts of leukocytes. We recorded the neutrophils and the lymphocytes counts, and calculated the neutrophil to lymphocyte ratio (*Forget et al., 2013*).

The CRP was determined by turbidimetry (UniCel® DxC 800; Beckman Coulter, Pasadena, California, U.S.A.) on a serum or plasma sample. During the reaction, a particle coated with anti-CRP antibody binds to the CRP in the patient sample, forming insoluble aggregates. The system monitors the change in absorbance at 600 nm. This change in absorbance is proportional to the concentration of CRP in the sample and is used by the system to calculate and express the concentration of CRP in a nonlinear calibration curve

to a single point, adjusted and predetermined. A value of CRP < 1.0 mg/dL was considered as normal.

## Statistical analysis

We compared patients with and without post-operative complications using a Chi-square for categorical variables and a (paired) Student $t$-test for continuous ones as appropriate. Uni- and multivariate analyses using logistic/linear regression models (with stepwise backward regression for multivariate analysis) were used. $P < 0.05$ was considered statistically significant. Data are expressed as mean ($\pm$ sd), mean [95% confidence interval], or number (percentage).

STATISTICA (data analysis software system) version 7 (Statsoft, Inc., 2004) was used for all analyses.

# RESULTS

## Baseline characteristics, procedures and postoperative complications

Preoperative characteristics, type and duration of procedures are detailed in Table 1.

During the postoperative period, 45 patients presented 69 complications. Two patients died from septic shock 20 days and 33 days after surgery of a failed anastomosis (Table 2). Patients with complications were comparable in term of preoperative characteristics, type and duration of procedure ($P > 0.05$) (data not shown).

## Inflammatory response after abdominal surgery

The NLR increased at $D + 1$ and, on average, returned to baseline at $D + 7$ unless complications (Table 3 and Fig. 1) ($P < 0.05$). Preoperative NLR is not significantly associated with postoperative complications whereas it is the case of NLR at $D + 7$. CRP presents a delayed peak compared to the NLR, increasing at $D + 2$ and not normalizing at $D + 7$, either there were complications or not (Fig. 1).

### Univariate analysis

To detect potential confounders, we performed a univariate analysis using a general logistic/linear regression model to observe the possible associations between postoperative complications and preoperative variables. When investigating inflammatory markers, the NLR at $D + 7$ was associated with more complications ($P < 0.0001$) (Table 4).

### Multivariate analysis

When including these variables into a multivariate analysis, and using a general logistic/linear regression model with stepwise backward regression, the NLR value at $D + 7$ was the only variable independently associated with more complications ($P < 0.001$), whereas none of the other factors, including CRP, showed any independent correlation. The American Society of Anesthesiologists Score (ASA) was not included because of the high risk of colinearity of this variable with other comorbidities, precluding its inclusion in a multivariate model. A performance analysis showed an area-under-the receiver operating curve (AUC) of 68% (95% CI [56–81]) ($P = 0.006$). The optimal cut-off of the NLR to predict

**Table 1 Preoperative characteristics, incidence of chronic diseases, type and duration of surgery and anesthesia, use of epidural analgesia.** Data are represented as median (range), mean ± SD or number (%). Two patients had 2 types of surgery.

| Baseline characteristics | |
| --- | --- |
| Age (years) | 62 (27–80) |
| Weight (kg) | 69.5 ± 15.5 |
| Height (cm) | 169.5 ± 9.0 |
| Sex (female/male) | 32/50 (39.0/60.9) |
| American Society of Anesthesiologists Score 2/3 | 44 (53.6)/ 38 (46.3) |
| Chronic diseases | |
|     Cirrhosis | 3 (3.6) |
|     Chronic obstructive pulmonary disease | 4 (4.8) |
|     Hypertension | 31 (37.8) |
|     Peripheral vascular disease | 14 (17.0) |
|     Coronary artery disease | 7 (8.5) |
|     Other cardiomyopathy | 6 (7.3) |
|     Diabetes mellitus | 6 (7.3) |
| Preoperative biological values | |
|     Hemoglobin (g dL$^{-1}$) | 12.6 ± 2.0 |
|     Serum creatinine (mg dL$^{-1}$) | 0.96 ± 0.3 |
| Type of surgery | |
|     Upper gastrointestinal | 12 (14.6) |
|     Hepato-biliary | 26 (31.7) |
|     Lower gastrointestinal | 46 (56.1) |
| Laparoscopic approach | 10 (12.2) |
| Duration of surgery (min) | 298 ± 139.5 |
| Duration of anesthesia (min) | 351 ± 141.5 |
| Epidural analgesia | 62 (75.6) |

occurrence of postoperative complications is 5.5, with a sensitivity of 66% and a specificity of 77%. The same analysis for the CRP did not show any significant result ($P > 0.05$).

## DISCUSSION

Complications are associated with a greater inflammatory response to abdominal surgery. This is better reflected by a significantly higher NLR, 7 days after the surgery, than by the CRP level. Indeed, the NLR remains elevated in patients developing post-operative complications compared with non-complicated outcome patients. In a multivariate analysis, the NLR value at $D + 7$ was the only factor associated with postoperative complications.

CRP levels have a delayed kinetics compared to the NLR, increasing at $D + 2$ and not normalizing at $D + 7$ either there are complications or not.

Our results extend previous findings showing that NLR is an independent marker of impaired outcome. In fact, the NLR has already been associated with morbidity or mortality of patients with cancer, cardiovascular disease, or chronic renal failure (*Forget et al., 2013*; *Proctor et al., 2011*; *Ding et al., 2010*; *Walsh et al., 2005*; *Kim et al., 2011*).

**Table 2 Postoperative outcome: complications, events and intensive care unit/hospital stay.** Data are represented as mean ± SD or number (%). Other infections than of surgery site concern: pulmonary, line-related, and other abdominal infection (like urologic infections). Cardiovascular complications concern: acute myocardial infarction, acute lung injury/acute respiratory distress syndrome, pulmonary edema, or arrhythmia.

| Postoperative outcome | | |
|---|---|---|
| Postoperative complications (N) | 69 | |
|     Infection of surgery site | | 16 (19.5) |
|     Other infections | | 13 (15.8) |
|     Cardiovascular complications | | 12 (14.6) |
|     Coagulopathy | | 11 (13.4) |
|     Upper digestive hemorrhage | | 7 (8.5) |
|     Leakage of anastomosis | | 10 (12.2) |
| Morbidity (event per patient) | 1.35 ± 2.0 | |
| Mortality | 2.0 (2.4) | |
| Length of stay | | |
|     Postoperative mechanical ventilation | 4 (4.9) | |
|     Intensive care unit (days) | 2.0 ± 6.5 | |
|     Hospital (days) | 15.6 ± 16.1 | |

**Table 3 C-reactive protein (CRP) and Neutrophil-to-Lymphocyte ratio (NLR) values in 82 patients: preoperatively (preop), at day + 1 (D1), day + 2 (D2) and day + 7 (D7), and presenting (or not) postoperative complications after major abdominal surgery.** Data are presented as mean ± SD. $P$ value <0.05 is considered significant.

| | Total (N = 82) | With complications (N = 45) | Without complication (N = 37) |
|---|---|---|---|
| CRP preop | 1.67 ± 3.74 | 1.98 ± 4.80 | 1.29 ± 1.76 |
|   CRP $D1$ | 9.17 ± 5.44 | 9.84 ± 5.71[***] | 8.34 ± 5.06[***] |
|   CRP $D2$ | 17.56 ± 8.89 | 19.84 ± 9.06[***] | 14.72 ± 7.91[* ***] |
|   CRP $D7$ | 8.13 ± 6.88 | 9.60 ± 7.74[***] | 6.32 ± 5.20[***] |
| NLR preop | 4.0 ± 4.91 | 3.89 ± 5.35 | 4.13 ± 4.43 |
|   NLR $D1$ | 14.43 ± 11.12 | 14.30 ± 11.37[***] | 14.58 ± 10.97[***] |
|   NLR $D2$ | 10.98 ± 7.07 | 11.87 ± 7.64[***] | 9.79 ± 6.13[***] |
|   NLR $D7$ | 7.96 ± 8.12 | 10.73 ± 9.86[***] | 4.73 ± 3.38[**] |

**Notes.**
[*] $P < 0.05$.
[**] $P < 0.001$ comparing the patients with and without complications (Student $t$-test).
[***] $P < 0.001$ compared with preoperative value (paired Student $t$-test).

Specifically, during the perioperative period, *Vaughan-Shaw, Rees & King (2012)* conducted a retrospective analysis of patients aged 80 years or over undergoing emergency abdominal surgery. Multivariate analysis identified NLR to be an independent predictor of 30-day outcome yet CRP did not predict outcome at any time point.

Taken together, this results show that NLR is a simple biomarker, widely available and probably more efficient than CRP, that can be used in various type of populations to

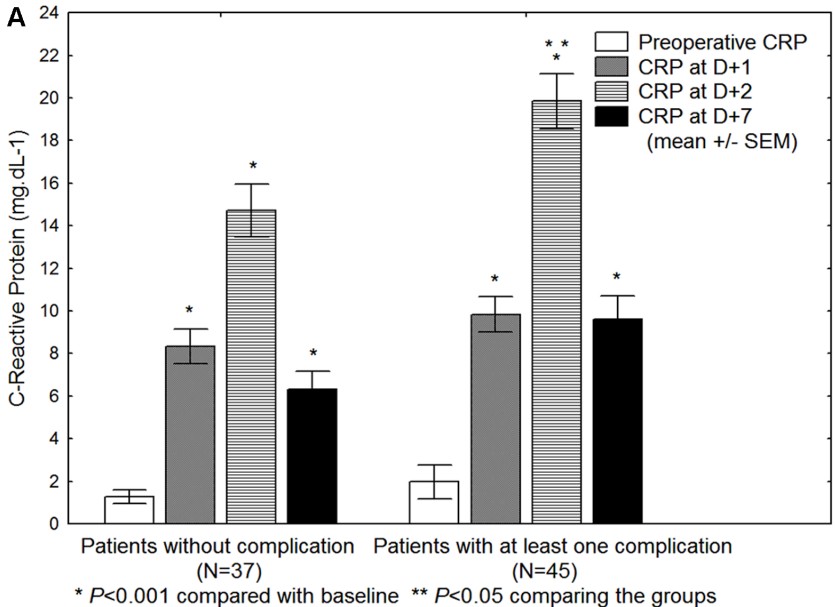

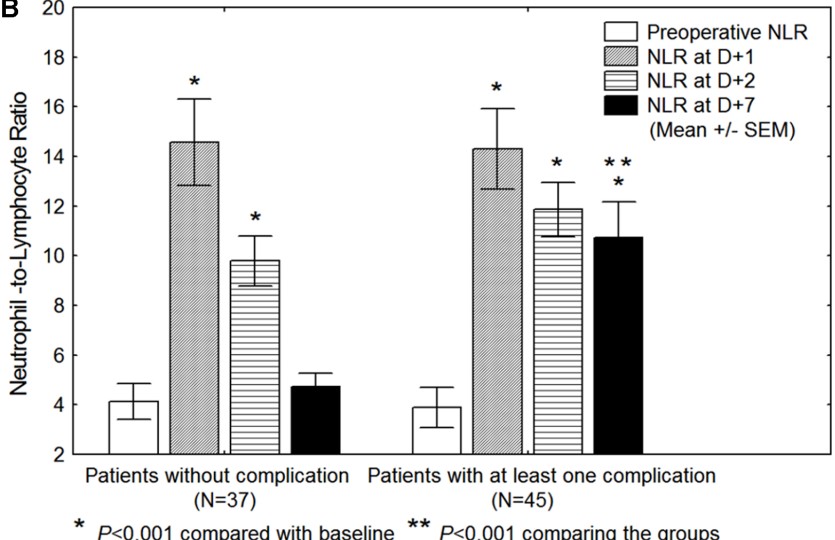

**Figure 1** **(A) C-reactive protein (CRP) and (B) Neutrophil-to-Lymphocyte ratio (NLR) values in 82 patients: preoperatively, at day + 1 (*D1*), day + 2 (*D2*) and day + 7 (*D7*), and presenting (or not) postoperative complications after major abdominal surgery.** Data are presented as mean ± SEM. *P*-value after (paired) Student *t*-tests.

monitor the inflammatory reaction and potentially able to help the clinician to perform early diagnoses of postoperative complications.

As our work was not designed for comparing techniques, further work may focus on the influence of various factors on the NLR such as the surgical technique (laparoscopy versus lapatomy) and anesthesia. Indeed, some potentially interesting factors were not considered in this work. This is a limit of this kind of post-hoc (retrospective) analysis. For example, the systemic IL-6 response is related to the magnitude of surgical trauma, as reflected by the

**Table 4 Uni- and multivariate analysis regarding risk factors of postoperative complications in 82 patients undergoing major abdominal surgery.** C-reactive protein (CRP) and Neutrophil-to-Lymphocyte ratio (NLR) values are considered at day + 1 (D1), day + 2 (D2) and day + 7 (D7).

| | Odds ratio | 95% confidence interval | | P |
|---|---|---|---|---|
| **Univariate analysis** | | | | |
| Cirrhosis | 1.66 | 1.08 | 2.56 | 0.02 |
| Peripheral vascular disease | 1.65 | 1.33 | 2.04 | <0.001 |
| Coronaropathy | 1.62 | 1.23 | 2.14 | <0.001 |
| CRP preop | 1.03 | 0.90 | 1.18 | 0.64 |
| CRP D1 | 0.96 | 0.90 | 1.03 | 0.27 |
| CRP D2 | 1.03 | 0.99 | 1.06 | 0.10 |
| CRP D7 | 0.97 | 0.93 | 1.01 | 0.15 |
| NLR preop | 0.97 | 0.87 | 1.08 | 0.59 |
| NLR D1 | 0.98 | 0.95 | 1.01 | 0.18 |
| NLR D2 | 1.02 | 0.97 | 1.07 | 0.37 |
| NLR D7 | 1.05 | 1.01 | 1.09 | 0.01 |
| **Multivariate analysis** | | | | |
| NLR D7 | 1.03 | 1.01 | 1.04 | <0.001 |

complexity of the surgical procedure and the duration of the abdominal operation (*Sido et al., 2004*). Regarding the influence of the anaesthesia, *Kim et al. (2011)* demonstrated that TIVA with propofol and remifentanil compared with inhalational anesthesia with sevoflurane could modify the leukocytic alterations, including neutrophil-to-lymphocyte ratio in peripheral blood during the postoperative period of laparoscopy-assisted vaginal hysterectomy, although the significance of these changes remains largely unanswered (*Carli & Annetta, 2003*). Finally, we have to recognize that we did not prove any clinical usefulness, which is questionable for a marker available seven days after surgery; earlier would be better. The CRP level, assessed two days after surgery, was a potential earlier marker. The possibility exists for considering both CRP and NLR values in a predictive score. However, although CRP is statistically different in patients with complications, it was not an independent predictor in multivariate analysis, challenged this approach.

## CONCLUSION

In this series of patients, the NLR at D + 7 is significantly associated with postoperative complications after major abdominal surgery and may be a simple but important biomarker, as this was not the case of CRP level.

### Funding

This work was exclusively supported by the Department of Anesthesiology of the Cliniques universitaires Saint-Luc, Brussels, Belgium. The funders had no role in study design, data collection and analysis, decision to publish, or preparation of the manuscript.

## Grant Disclosures

The following grant information was disclosed by the authors:
Department of Anesthesiology of the Cliniques universitaires Saint-Luc.

## Competing Interests

The authors declare there are no competing interests.

## Author Contributions

- Patrice Forget and Valérie Dinant conceived and designed the experiments, performed the experiments, analyzed the data, contributed reagents/materials/analysis tools, wrote the paper, prepared figures and/or tables, reviewed drafts of the paper.
- Marc De Kock conceived and designed the experiments, performed the experiments, wrote the paper, reviewed drafts of the paper.

## Human Ethics

The following information was supplied relating to ethical approvals (i.e., approving body and any reference numbers):
CEBHF of the Universite Catholique de Louvain. NCT00816153.

## Data Deposition

The following information was supplied regarding the deposition of related data:
DIAL-UCL.

## Supplemental Information

Supplemental information for this article can be found online at http://dx.doi.org/10.7717/peerj.713#supplemental-information.

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
