# Peer review of "Is the Neutrophil-to-Lymphocyte Ratio more correlated than C-reactive protein with postoperative complications after major abdominal surgery?"

_PeerJ, doi:10.7717/peerj.713_

## Round 0.1 · original submission · Major Revisions

· Academic Editor

Major Revisions

It seems to me that the reviewers like the overall idea and findings of the study but they require some more details/data or an expanded discussion - this is somehow good news since no new experiments have been requested. Please address the reviewer comments.

Reviewer 1 ·

Basic reporting

No Comments

Experimental design

The experimental design is correct.
But the authors should mention clearly that this study is not really prospective study and post-hoc analysis. This is one of the limitations in the study.

Validity of the findings

1. The authors just focused on high NLR on D+7. If so, they should discuss about cut-off value for the prediction of post-operative complications. Judging from the data in figure, the absence of decrease NLR from D+1 to D+7 seems to be more different between patients with complication and without complication. In the view of clinical usefulness, the earlier expected value is more useful for clinical use. They should discuss about these more.
2. From the data in table 2 and figure, D+2 CRP in patients with at least one complication seems to be higher than that in patients without complication. They should discuss the difference. If there is significant difference, higher CRP on D+2 can estimate post-operative complication earlier than high D+7 NLR.
3. They should also discuss about the combination of high CRP at D+2 and high NLR at D+7.

Additional comments

1. In table 1, the age should be expressed in median.
2. Table 2 should be improved because it is difficult to understand.
3. The authors should show the odds ratios including 95% confident interval for univariate and multivariate analyses.
4. In table 1, ASA score spelled out.

Reviewer 2 ·

Basic reporting

No Comments.

Experimental design

No Comments.

Validity of the findings

No Comments.

Additional comments

This theme is interesting and current topics. However, there are some questions mentioned below.

1) First of all, although the results of univariate and multivariate analyses were most important in this study, there were no data in the tables. The authors should demonstrate the details of such data.

2) Because there were several types of surgery in the study, the authors show the distribution of such procedures in the two groups (with/without complication). In addition, the authors should show that there were no significant differences in the heterogeneity of background characteristics between the two groups.

3) The authors should reveal the number of patients who had complications within 7 days after surgery, because preoperative systemic inflammatory response predicts postoperative complications.

---

## Round 0.2 · accepted · Accept

· Academic Editor

Accept

All the comments have been taken into account.